# Spatial and Temporal Volatility of PM2.5, PM10 and PM10-Bound B[a]P Concentrations and Assessment of the Exposure of the Population of Silesia in 2018–2021

**DOI:** 10.3390/ijerph20010138

**Published:** 2022-12-22

**Authors:** Dorota Kaleta, Barbara Kozielska

**Affiliations:** Department of Air Protection, Silesian University of Technology, 22B Konarskiego St., 44-100 Gliwice, Poland

**Keywords:** air pollution, particles, benzo[a]pyrene, seasonal variation, carcinogenic risk assessment, lifetime lung cancer risk

## Abstract

Air pollution both indoors and outdoors is a major cause of various diseases and premature deaths. Negative health effects are more frequently observed in a number of European countries characterized by significant pollution. In Poland, especially in Upper Silesia, the most serious problem is the high concentration of particulate matter (PM) and PM10-bound benzo[a]pyrene (B[a]P). The main source of these two pollutants is so-called “low emissions” associated with the burning of solid fuels mainly in domestic boilers and liquid fuels in road traffic. This study examined the variability in the PM and PM10-bound B[a]P concentrations and their relationships with meteorological parameters, i.e., atmospheric pressure, air temperature and wind speed, in 2018–2021 at 11 monitoring stations. In many Silesian cities, the average annual concentrations of PM10, PM2.5 and B[a]P were much higher than those recorded in other European countries. At each station, the average daily PM10 concentrations were exceeded on 12 to 126 days a year. Taking into account the WHO recommendation for PM2.5, the highest recorded average daily concentration exceeded the permissible level by almost 40 times. The same relationships were observed in all measurement years: PM10 concentrations were negatively correlated with air temperature (R = −0.386) and wind speed (R = −0.614). The highest concentrations were observed in the temperature range from −15 °C to −5 °C, when the wind speed did not exceed 0.5 m·s^−1^. The calculated lifetime cancer risk (LCR) associated with the exposure to B[a]P in the Silesian Voivodeship suggested 30–429 cases per 1 million people in the heating season depending on the scenario used for the calculations (IRIS, EPA or WHO).

## 1. Introduction

Air pollution occurring indoors and outdoors is the most serious environmental health threat in the modern world [1]. The pollutants with the most serious impacts on human health are particulate matter, nitrogen dioxide and ground-level ozone. Their synergistic effect is particularly dangerous [2,3]. They are the main causes of illness and premature death (in 2016, it was estimated that air pollution was the cause of 4.2 million premature deaths [1]) caused by cardiovascular diseases, i.e., heart diseases and strokes, as well as respiratory diseases, predominantly lung diseases and lung cancer [4,5]. In the countries of the European Union (EU), approx. 300,000 premature deaths caused by fine particulate matter were recorded for the year of 2019 [6,7]. It is worth emphasizing that the onset of symptoms of the diseases caused by air pollution is delayed after exposure.

Research shows that the environmental burden of disease varies across Europe. In areas with significant environmental pollution, negative health effects are seen more often. Exceedances of the daily limit value recommended by the EU for PM10 (50 µg·m^−3^) occur throughout the continent; however, the highest concentrations are observed in Northern Italy, Croatia, Bulgaria, Serbia, Kosovo, Turkey, Bosnia and Herzegovina, North Macedonia and Poland [8,9,10,11]. Unfortunately, Poland is considered to be a country with some of the severest problems related to air pollution in the EU, mainly related to PM2.5, PM10 and benzo[a]pyrene pollutants. Both particulate matter and benzo[a]pyrene are classified by the International Agency for Research on Cancer (IARC) as carcinogenic to humans (Group 1) [12]. It should be noted that, for many years, B[a]P has been recognized as a good and sufficient indicator of human exposure to priority polycyclic aromatic hydrocarbons (PAHs) present in the air [13]. Studies conducted on animals confirmed that B[a]P can cause various types of neoplasms, such as gastrointestinal tract, liver, kidney, respiratory tract, pharynx and skin cancers [14]. In many countries, including Poland, the PM10-bound B[a]P concentration in the air is routinely monitored in order to ensure that its average annual concentration does not exceed the limit value (the established limit that the average annual concentration of B[a]P should not exceed is 1 ng·m^−3^ in most areas [15], and, according to the WHO recommendations, it is 0.12 ng·m^−3^ [16]). Over the last few decades, the interest in exposure to PM10-bound B[a]P and PM2.5-bound B[a]P has increased because they are associated with a broad range of health effects that have a major impact on public health.

In Poland, especially in Silesia, the highest share of the emissions of PM10, PM2.5 and B[a]P (77%, 87.9% and 97.8%, respectively) is attributed to communal and household sources [17]. Consequently, municipal emissions attributed to the combustion of fuels in domestic boilers (mainly coal and wood and, despite the ban [18], municipal waste [19,20,21]) are considered the main cause of exceeding the concentration limits of PM and B[a]P in the air in Poland. The evidence for this is that the PM and B[a]P concentrations are higher in the heating season than in the non-heating season [22,23,24]. According to the most recent available data, in 2018, almost half of the households in Poland, i.e., 45.4%, still used solid-fuel heating devices [25]. Other important sources of emissions include industrial activities [26], as well as agricultural and road transport, especially in urban areas [27,28]. According to the ranking prepared by the World Health Organization (WHO), 36 out of 50 cities with the worst air quality in the EU are located in Poland, most of them in Upper Silesia [29].

During the heating season in Silesia, so-called smog incidents [30] are also more frequent, during which the concentration of pollutants increases above the permissible standards. The formation of smog depends on the coexistence of a number of meteorological factors, i.e., low temperature, a low wind speed, thermal inversion and the height of the mixing layer. The occurrence of smog in Poland is primarily associated with excessive air pollution, especially the particulate matter of large particles (PM10 and PM5), the particulate matter of medium and small particles (PM2.5 and PM1) and polycyclic aromatic hydrocarbons (PAHs), and, to a lesser extent, sulfur oxides, nitrogen oxides and ozone or carbon monoxide. According to a recent study [31], Poland can be divided into three smog regions (Figure 1): Region I comprises the south and a part of Silesia, where smog incidents occur frequently and last the longest. This region includes cities and poorly ventilated areas, where a stable atmosphere with temperature inversion occurs frequently. Region II is a strip of land stretching from the Lublin Upland to the Wielkopolska Lowland, with a lower frequency of smog incidents. Finally, Region III comprises northern Poland, where smog incidents are the least frequent, which is attributed to the unstable atmosphere dominant in this area. Analyses of smog incidents in Silesia have shown significant increases in the number of registered patients for ambulatory care due to bronchitis and asthma exacerbation in response to increases in PM2.5 concentrations [32,33]. According to [32], the described dependencies are visible during the first few days of smog episodes, and they do not cease in the following days. In the case of hospitalization due to acute respiratory diseases, a greater number of registered events appeared with a two-day delay from the observation of a smog incident. During the periods of smog incidents, a significant increase in mortality was also observed [32].

Although various legal regulations and programs have been introduced in Poland [34,35,36], in the era of the energy crisis and increasing energy poverty [37,38], we can expect a deterioration in air quality in the near future. It should also be remembered that the quality of outdoor air also has a significant impact on the quality of indoor air [39], especially in places where natural ventilation is used. It has been observed that higher concentrations of pollutants characteristic of outdoor air occur inside buildings where windows and doors face directly onto streets, main communication routes or car parks [40,41]. The aim of this study was to examine the correlation between the concentrations of PM10, PM2.5, B[a]P and meteorological parameters and to analyze the risk of developing cancer resulting from inhalation exposure to B[a]P among the inhabitants of Upper Silesia (Poland).

## 2. Materials and Methods

### 2.1. Study Area

Silesian Voivodeship is a province located in the southern part of Poland, and it occupies 12,333 km^2^, with a population of about 4.47 million (363 inhabitants per one square kilometer [42]). It is one of the most urbanized and industrialized regions of Central Europe. There are 30 monitoring stations located in this region [17]. In the frame of this research, the daily concentrations of particulate matter PM2.5, PM10 and PM10-bound B[a]P were analyzed in all monitoring stations; however, only 11 localizations (Figure 2), in which PM and B[a]P were tested simultaneously in a given time period, were chosen for further analyses (Appendix A). The locations of all stations fulfilled the conditions required for the so-called urban background site (UB). PM2.5 was initially only measured at 3 measuring stations, and from 2021, it was measured at 5 stations. Only those stations for which a measurement coverage of 80% was ensured each year were included in further analyses. The heating season and the non-heating season were analyzed separately. It was assumed that the heating season lasts from the beginning of October to the end of March each year. As a criterion for a particular day qualifying as a smog incident day, the exceedance of the average daily PM10 concentration of 50 μg·m^−3^ was assumed, in accordance with the literature [43] (the 24 h threshold value).

### 2.2. Sample Collection

The available data concerning 24 h PM2.5, PM10 and PM10-bound B[a]P ambient concentrations in the Silesian Voivodeship from the period of 2018–2021 were taken from the Chief Inspectorate of Environmental Protection (CIEP) website [44].

At the selected monitoring stations, suspended particulate matter was manually sampled for 24 h a day, using high-volume samplers with an air flow of 30 m^3^·h^−1^. Samplers were equipped with a separate PM2.5 or PM10 head and a quartz fiber filter container, introducing subsequent filters into the measurement duct automatically. The sampled dust mass was determined with the use of the reference gravimetric method according to the PN-EN 12341:2014-07 standard [45]. The detection limit of this method is 1 μg·m^−3^.

PM10-bound B[a]P was extracted with an organic solvent and then analyzed using high-performance liquid chromatography (HPLC) with a fluorescence detector or gas chromatography with a mass spectrometry detector (GC-MS) in accordance with the PN-EN 15549:2011 standard [46]. These methods are applicable to the measurement of B[a]P in the concentration range of 0.04 to 20 ng·m^−3^, and their detection limits are below 0.04 ng·m^−3^ [46].

### 2.3. Statistical Analysis

In order to present a concise characterization of the analyzed concentrations, the basic parameters of descriptive statistics were calculated for each station, i.e., mean, median, standard deviation, maximum and minimum values, and the 1st and 3rd quartiles. The Spearman’s rank correlation coefficient was used to analyze the data on PM10 and B[a]P concentrations due to the fact that the criterion of similarity between the distribution of the examined variables and the normal distribution was not met. The similarity of the distribution of the analyzed variables to the normal distribution was examined using the Shapiro–Wilk test. All analyses were performed using Statistica 13 software, with a significance level of 0.05.

### 2.4. Lifetime Cancer Risk Assessment

Benzo[a]pyrene is recognized as a marker of the carcinogenic potential of a mixture of polycyclic aromatic hydrocarbons [47]; for this reason, it was taken as the indicator of PAH carcinogenic risk in this study similarly to other research [48,49].

Carcinogenic risk is often associated with the toxicity equivalence factor (TEF) of each of the 16 PAHs compared to B[a]P [50,51,52,53,54]. The B[a]P international toxic equivalent (TEQ) concentration is calculated by multiplying the concentration of each PAH with its corresponding TEF. As this study focused on B[a]P, it only took the TEQ of B[a]P with its TEF as a unity into consideration. Furthermore, the inhalation unit risk for benzo[a]pyrene was derived with the intention that it would be paired with EPA’s relative potency factors for the assessment of the carcinogenicity of the PAH mixtures [55].

The lifetime cancer risk (LCR) from the inhalation exposure to B[a]P can be expressed as follows:*LCR* = *TEQ_B[a]P_* × *IUR_B[a]P_*(1)
where:

*TEQ_B[a]P_*—B[a]P equivalent equal to the B[a]P concentration in this study, µg·m^−3^;

*IUR_B[a]P_*—inhalation cancer unit risk factor of exposure to B[a]P, *IUR* = 6 × 10^−4^ (µg·m^−3^)^−1^.

This exposure duration scenario included full lifetime exposure (assuming a 70-year lifespan). The inhalation unit risk (IUR) is defined as the upper-bound excess lifetime cancer risk estimated to result from continuous exposure to an agent at a concentration of 1 µg·m^−3^ in the air [56]. The IUR_B[a]P_ values for the carcinogenic benzo[a]pyrene used in this study were extracted from the database provided by the Integrated Risk Information System (IRIS) [57,58]. Previously, according to the Office of Environmental Health Hazards Assessment of the California Environmental Protection Agency, IUR_B[a]P_ was taken as 1.1 × 10^−6^ (ng·m^−3^)^−1^ [59,60]; however, the WHO suggests an IUR_B[a]P_ value of 8.7 × 10^−6^ (ng·m^−3^)^−1^ [61].

## 3. Results and Discussion

Excessive air pollution in Poland has been present for decades, especially in Upper Silesia. Studies of total particulate matter, conducted as early as the 1970s, showed that, in most cities in this area, the average concentrations of TSP exceeded 300 μg·m^−3^ annually, and the B[a]P concentrations were higher than 200 ng·m^−3^ [62,63]. The currently monitored concentrations of PM2.5, PM10 and B[a]P have decreased compared to the last century, but they are still high [44]. The highest emissions of dust pollutants from particularly burdensome industrial plants in Poland have been recorded in the area of the Silesian Voivodeship for many years. In 2020, these amounted to 4.4 thousand tone in total [64].

### 3.1. Concentrations of PM2.5 and PM10

The PM2.5 and PM10 concentrations measured in the period of 2018–2021 at 11 monitoring stations located in the Silesian Voivodeship were characterized by some variability (Appendix A). The average annual concentrations, depending on the location and year, ranged as follows: from 19.62 to 38.55 μg·m^−3^ for PM2.5 and from 22.87 to 54.88 µg·m^−3^ for particulate matter PM10. These concentrations are much higher than the concentrations recorded in other European countries [29,65]. This means that, in many Silesian cities, the average annual concentrations exceeded the Polish standards of permissible concentrations (20 μg·m^−3^ for PM2.5 and 40 μg·m^−3^ for PM10). In comparison with the WHO guidelines (5 μg·m^−3^ for PM2.5 and 15 μg·m^−3^ for PM10), the average annual concentrations exceeded the safe level at every station. The highest concentrations were observed in the cities of Rybnik, Żywiec and Godów, where the share of the so-called low emissions in the overall balance is high [66].

In relation to the permissible limits regulated by Polish law and recommended by the WHO, the average daily concentrations of PM2.5 and PM10 were exceeded at almost every station (Figure 3 and Figure 4). According to the standards, the average daily concentration of PM10 may be exceeded for only 35 days a year; however, at most of the measuring stations, the daily average concentrations were exceeded from 12 to 126 days a year. A great number of these exceedances (approx. 73% of the total number of days) took place in the heating season (Appendix A). The highest concentration of PM2.5 was noted in the city of Godów in 2019, where it amounted to 228.60 μg·m^−3^, and the recommended level was exceeded almost 40 times according to the WHO guidelines (15 μg·m^−3^). Comparable daily concentration levels have only been recorded in some European cities (Sarajevo [67] and Belgrade [68]). These high concentrations and numerous exceedances of permissible limits are the cause of the frequent smog incidents in Silesia.

Due to the negative health effects of PM2.5, the ratio between the mass concentrations of PM2.5 and PM10, i.e., PM2.5/PM10, was additionally calculated (Table 1). For the entire calendar year, the PM2.5/PM10 ratio ranged from 0.6533 to 0.8804. In the heating and the non-heating seasons, the PM2.5/PM10 ratios were very similar, ranging from 0.6573 to 0.9877 and from 0.6451 to 0.7557, respectively, and they are similar to those in the literature [69,70]. For example, in Melpitz, located in eastern Germany, the average PM2.5/PM10 ratios were 0.72 in the summer and 0.82 in the winter [71]. The maximum difference between the heating and non-heating seasons was around 0.15. The calculations show that, in Silesia, the fine-grained PM fraction clearly predominates over the course-grained PM fraction. The increase in the emissions of fine and sub-micron particles is driven by the intensification of various activities, such as home heating based on coal fuel, industrial fuel combustion processes and the low mixing height of the air, resulting in the accumulation of secondary organic aerosol precursors and re-emissions [70,71].

### 3.2. Relationship between Some Meteorological Parameters and Particulate Matter Concentrations

The PM10 concentration depends on many environmental and anthropogenic factors. Meteorological conditions, i.e., wind direction and speed, air temperature, atmospheric pressure, precipitation, humidity, solar radiation, atmospheric stability and the height of the mixing layer, are considered to be factors of key importance [72,73,74]. They affect the diffusion, deposition and dilution of PM and, thus, the spatial distribution of pollutants in the bottom part of the atmosphere. In this paper, the authors analyzed selected meteorological parameters (i.e., wind speed, air temperature and atmospheric pressure) that were measured at the station in Rybnik. They were juxtaposed with the high concentrations that occurred regularly during the heating season at this station.

For this purpose, Spearman’s correlation coefficients were calculated (Table 2). The same dependencies were observed in all analyzed years. The PM10 concentrations were negatively correlated with air temperature (R = −0.386). Higher air temperatures lead to an effective vertical dispersion of pollutants, which results in low concentrations of particulate matter [72,75]. At low air temperatures, the activity of heating systems increases, and the height of the planetary boundary layer (PBL) [76] decreases, which causes a sharp increase in PM10 concentrations [77,78]. The speed of the wind was shown to be a very important factor, as it was strongly and negatively correlated with PM10 (R = −0.614). The highest concentrations of PM10 occurred in windless or almost windless conditions, which favors the formation of temperature inversion in the PBL [76], and this may cause an additional accumulation of pollutants in the ground level of the troposphere [79].

The highest concentration of PM10 was observed in a temperature range from −15 °C to −5 °C, when the wind speed did not exceed 0.5 m·s^−1^ (Figure 5). Similar results were observed in the literature [72,73,78,80]. The lowest average temperature (4.23 °C) in the heating season was recorded in Rybnik in 2018. Moreover, in the period from March 1 to March 6, the daily concentrations of PM10 and B[a]P remained at very high levels, exceeding the alarm level (150 µg·m^−3^ for PM10 according to [81]). Cyclic high concentrations occurred during every period characterized by very low temperatures and very low wind speeds (Figure 6).

### 3.3. Concentration of PM10-Bound B[a]P

In the Silesian Voivodeship in 2018–2021, the average concentration of PM10-bound B[a]P ranged from 6.2 to 7.1 ng·m^−3^ (Appendix A). The highest B[a]P concentrations were recorded in Rybnik; in the 2018, 2019, 2020 and 2021 heating seasons, the concentrations were 23.5, 27.7, 15.9 and 15.7 ng·m^−3^, and in the non-heating seasons, the concentrations were 0.8, 2.9, 2.2 and 1.3 ng·m^−3^, respectively. In Częstochowa, the average annual concentrations of B[a]P in the studied period were the lowest, but they were still three times higher than the permissible values (EU). Despite this, the recorded concentration of B[a]P in Częstochowa in 2018 was the lowest in the entire voivodship, but the permissible concentrations were still exceeded as many as 20.6 times according to the WHO recommendation level of 0.12 ng·m^−3^. The observed PM10-bound B[a]P levels showed a clear seasonal pattern (Figure 7), with the highest values recorded during the cold season (heating season), which was due to fuel burning in local households, as well as low wind speed and air temperature, a low planetary boundary layer height and a stable atmosphere conducive to the accumulation of pollutants. In the Silesian Voivodship, the PM10-bound B[a]P concentrations of 4.0 to 27.7 ng were recorded from October to the end of March. Earlier studies conducted by Kozielska et al. [22] also indicated high concentrations of B[a]P in the 2009–2010 heating season in Silesia, and they are comparable to those presented in this work (Table 3). Significantly higher concentrations of B[a]P have only been found in India (Rajpur—43–76 ng·m^−3^), Pakistan (Lahore—81 ng·m^−3^) and China (Tangshan—61.6 ng·m^−3^) (Table 3). It should also be noted that, in the heating season, there may be as much as 380 μg B[a]P per 1 g of PM10, and, for example, in a coking plant, these values are in the range of 6.7 to 3356 μg·g^−1^ [82].

The possible emission sources around individual measuring stations can be indicated by analyzing the Spearman’s linear rank correlation coefficient between PM10 and B[a]P. The Spearman’s rank correlation coefficients and their corresponding levels of significance are presented in Table 4. Very strong positive correlations (R = 0.843 and R = 0.873) were found for Rybnik and Żywiec, thus suggesting that B[a]P comes from the same source of pollution, such as the combustion of solid fuels. Significant but slightly lower correlations (R = 0.723 and R = 0.721) between PM10 and B[a]P were observed in urban areas with high anthropogenic activity, especially in urban areas with elevated traffic levels (Katowice, Częstochowa). The obtained results suggest that, at the measuring stations exposed to the strong impact of the so-called low emissions from the combustion of solid fuels, the correlation between PM10 and B[a]P is higher than that at the sites exposed to both traffic emissions and fuel combustion in local furnaces.

### 3.4. Health Risk Assessment

The lifetime cancer risk (LCR) of inhalation exposure was estimated on the basis of the concentration of PM10-bound B[a]P in the atmospheric air in the Silesian Voivodeship. The lifetime lung cancer risk values calculated using IRIS *IUR_B[a]P_* are shown in (Table 5). The WHO estimated an acceptable LCR of 1 × 10^−6^ (1 in 1,000,000) to 1 × 10^−4^ (1 in 10,000) for carcinogens. Under most guidelines used for LCR assessment, an LCR between 1 × 10^−6^ and 1 × 10^−4^ suggests a potential cancer risk, while a potential risk is high at LCR > 10^−4^ [15]. The LCR values were calculated for three scenarios: (a) annual, (b) the heating season and (c) the non-heating season. In the area of the Silesian Voivodeship, the LCR of the inhalation exposure to B[a]P was at an acceptable level. The lowest risk was found in the non-heating season, with values in the range of 0.32 × 10^−6^ (Bielsko-Biała) to 1.08 × 10^−6^ (Rybnik). For the heating season, these values were in the range of 2.8 × 10^−6^ to 12 × 10^−6^. The highest LCR occurred in Rybnik, where the average concentration of B[a]P in the heating seasons was at a level of 20 ng·m^−3^ (15.6–24.7 ng·m^−3^). Widziewicz et al. [95] analyzed B[a]P concentrations in Poland in the period between 2010 and 2015 using inhalation cancer unit risk factors recommended by EPA and the WHO. According to the authors, LCR was above the acceptable limits, and it was 7.33 × 10^−4^ in most conservative scenarios (Opolskie Voivodeship). If we take into account the same conservative scenario as Widziewicz et al. [95] (WHO), then during the heating season in Godów, Rybnik and Żywiec, the LCRs were also above the acceptable limits at 1.16 × 10^−4^, 1.73 × 10^−4^ and 1.25 × 10^−4^, respectively [95].

Applying the suggested IRIS *IUR_B[a]P_* for lifetime (70 years) B[a]P exposure, the corresponding lifetime cancer risk is 3.88 × 10^−6^ on average while by WHO *IUR_B[a]P_* is 56.33 × 10^−6^ for the measurements in The Silesian Voivodeship (2018–2021) (Table 6). Thus if 1,000,000 people are exposed to 6.5 ng·m^−3^ of ambient B[a]P for 70 years, then lung cancer may develop in 17–248 people on average, depending on the adopted *IUR_B[a]P_* values (Table 6). If 1,000,000 people are exposed to 11.2 ng·m^−3^ (heating seasons) of ambient B[a]P, then lung cancer may develop in 30,429 people. 

The presented results for the cancer risk of the inhalation exposure to B[a]P are understated and probably underestimate the carcinogenic potential of airborne PAH mixtures. For example, in the research conducted by Kozielska et al. [22,23], it was found that, in urban areas in the Silesian Voivodeship, the share of B[a]P in the total of 16 PAHs was about 9% (6–12%) on average, and for the B[a]P equivalent (TEQ) calculated by Nisbet and LaGoy [50], the value was between 56 and 68%. In [88], the share of B[a]P in TEQ was 58–62% in Indian semi-urban and urban sites. The estimation of cancer risk also has a significant impact on the toxicity equivalence factor scheme [96]. Moreover, there are additional factors contributing to carcinogenicity. One of the complicating factors is that PAHs in the air are bound to particles that may cause adverse health effects themselves. The carcinogenic potential of PAHs may even be enhanced when combined with those particles [91]. Additionally, in particles of different particle fractions (PM10, PM2.5 and PM1), the share of individual PAHs can vary significantly [97,98,99]; therefore, it can be expected that carcinogenic potency also differs substantially for those particle fractions [28,91].

## 4. Conclusions

Currently, the quality of atmospheric air is one of the most important environmental problems. This especially concerns developed and developing countries, including Poland. Unfortunately, for years, Poland has occupied one of the first places in the classification of countries with the most polluted air in Europe, and air quality standards are exceeded in its dominant area, especially when it comes to particulate matter and B[a]P. As many as 36 cities with the worst air quality in Europe are located in Poland, and most of them are located in Upper Silesia. Considering the fact that particulate matter is characterized by different particle sizes and chemical compositions, depending on the place of occurrence and the season, it is the most serious health hazard. In the Silesian Voivodeship in 2018–2021, the concentrations of PM and PM10-bound B[a]P very often exceeded the permissible concentrations regulated by national and European regulations, and they were higher than the concentrations recorded in other European cities. Considering the WHO guidelines for PM, the average daily concentrations of PM2.5 and PM10 were exceeded for almost the entire heating season. The average number of exceedances per year was 210 for PM2.5 and 75 for PM10, respectively. In Silesia, the fine-grained PM fraction clearly predominated over the coarse-grained PM fraction (PM2.5/PM10 ratio was in the range of 0.6533 to 0.8804). High concentrations of PM were strongly negatively correlated with wind speed and air temperature. The highest temporary PM concentrations reaching over 800 μg·m^−3^ were observed in the temperature range of −15 °C to −5 °C when the wind speed did not exceed 0.5 m·s^−1^. Moreover, the daily concentrations remained at a very high level at that time, exceeding the alarm level (150 µg·m^−3^ for PM10).

The concentrations of B[a]P recognized as a marker of the carcinogenic potential of a PAH mixture ranged from 4 ng·m^−3^ to around 28 ng·m^−3^ from October to the end of March each year, while EU regulations recommend a value of 1 ng·m^−3^, and the WHO recommends a value of 0.12 ng·m^−3^. Importantly, even 1 g of PM10 can contain 380 μg B[a]P.

According to IRIS, the estimated lifetime cancer risk (LCR) due to the inhalation exposure to PM10-related B[a]P concentrations in Silesia is acceptable. If we take into account the WHO scenario, then in Godów, Rybnik and Żywiec, the LCRs are above the acceptable limits during the heating season. The calculated lifetime lung cancer risk associated with exposure to B[a]P in the measured heating seasons in the Silesian Voivodeship suggested 30–429 cases per 1 million people depending on the IRIS, EPA or WHO scenario used.

## Figures and Tables

**Figure 1 ijerph-20-00138-f001:**
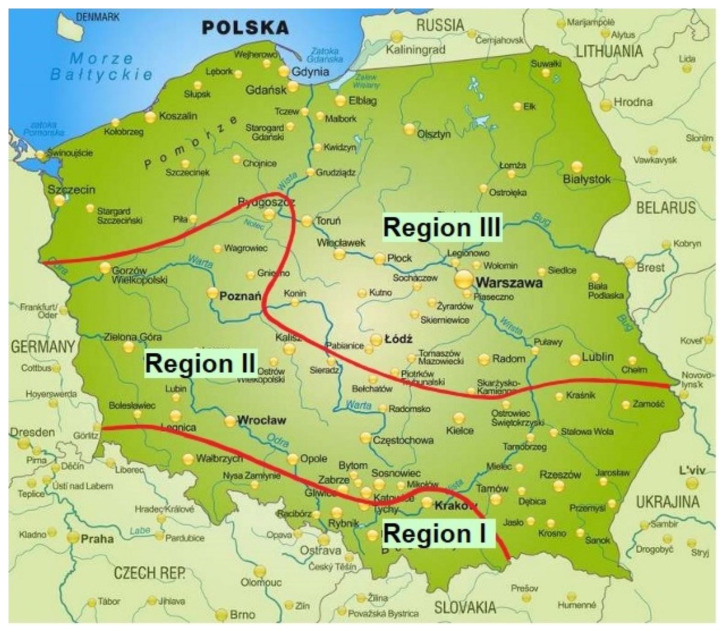
“Smoggy” areas with higher PM concentrations in air in Poland [31]. **Region I**—smog incidents occur frequently and last the longest; **Region II**—smog incidents occur with lower frequency; **Region III**—smog incidents occur rarely.

**Figure 2 ijerph-20-00138-f002:**
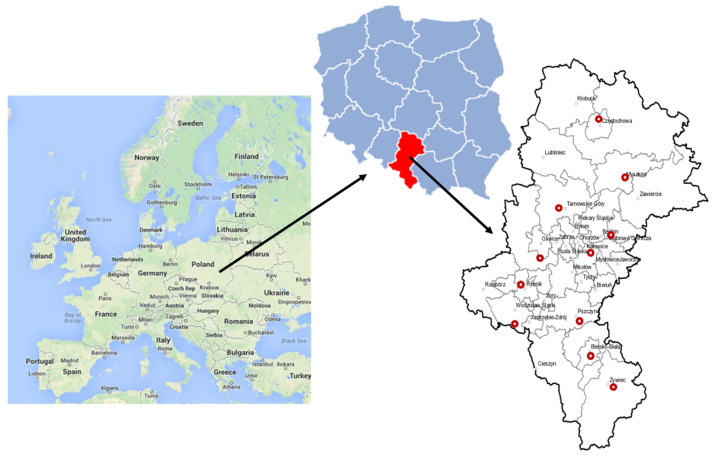
Location of selected air monitoring stations in Silesian Voivodeship.

**Figure 3 ijerph-20-00138-f003:**
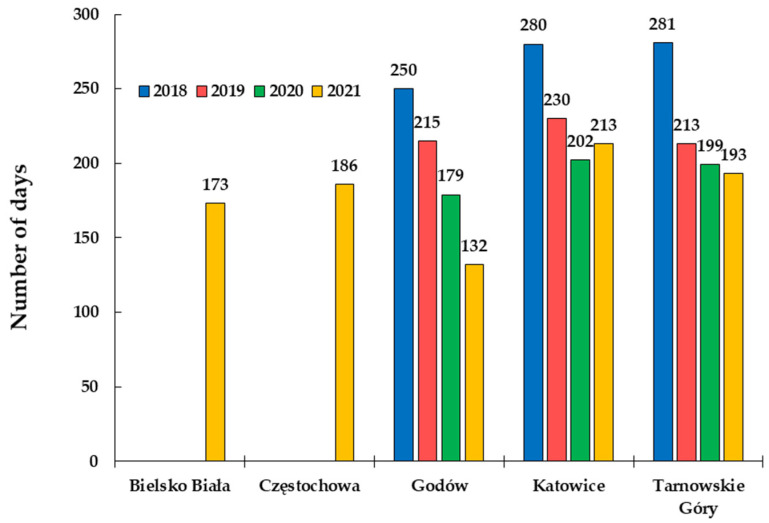
Number of days exceeding the values of the WHO-recommended daily mean concentrations of PM2.5.

**Figure 4 ijerph-20-00138-f004:**
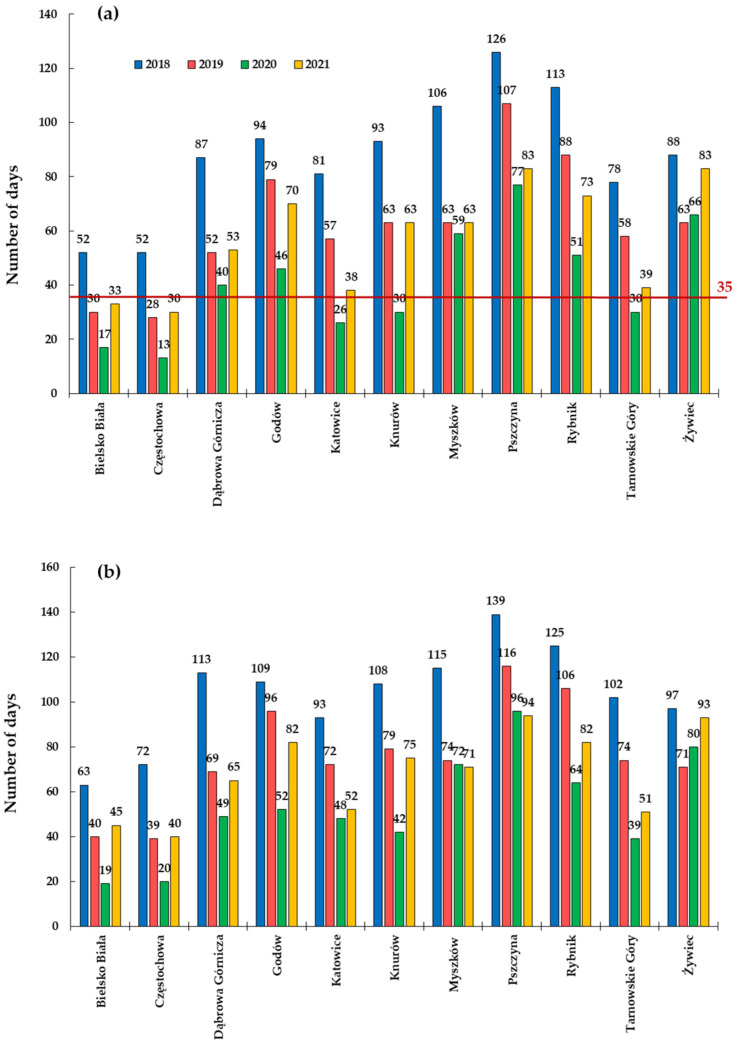
Number of days exceeding the values of the recommended daily average concentrations of PM10 (**a**) according to national standards and (**b**) according to the WHO in Poland.

**Figure 5 ijerph-20-00138-f005:**
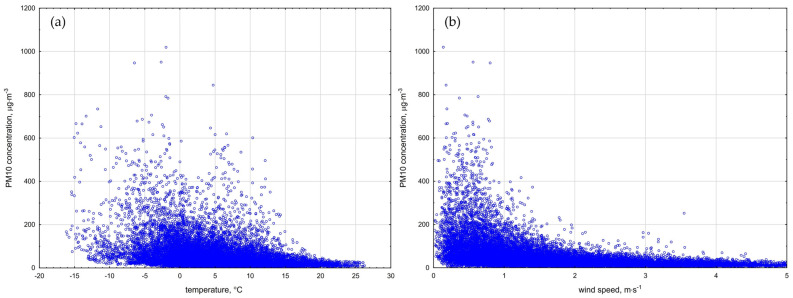
Changes in PM10 concentration depending on the (**a**) air temperature and (**b**) wind speed that occurred in the 2018 heating season in Rybnik.

**Figure 6 ijerph-20-00138-f006:**
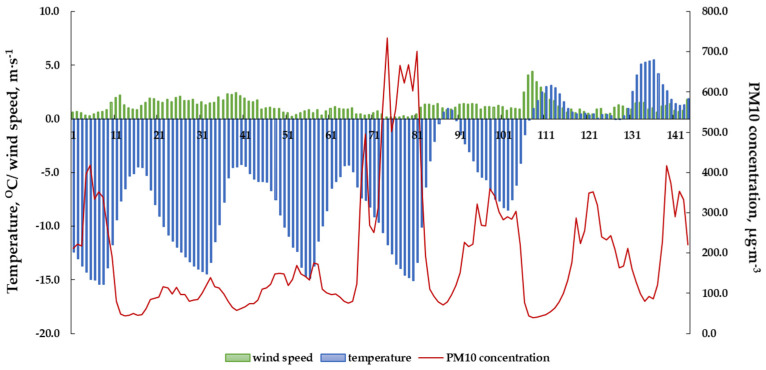
Distribution of 1 h PM10 concentrations and recorded meteorological parameters from 1–6 March 2018 in Rybnik.

**Figure 7 ijerph-20-00138-f007:**
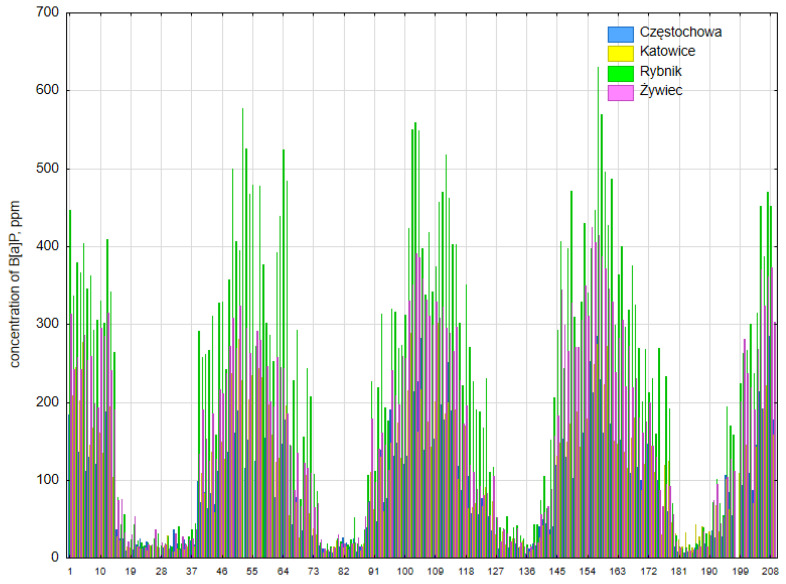
Seasonal variability of PM-bound B[a]P in individual weeks in 2018–2021.

**Table 1 ijerph-20-00138-t001:** The PM2.5/PM10 ratio in the period of 2018–2021 by heating and non-heating seasons.

	Godów	Katowice	Tarnowskie Góry	Bielsko Biała	Częstochowa
**2018**					
year	0.7453	0.7210	0.7235	-	-
heating season	0.7999	0.8024	0.7629	-	-
non-heating season	0.6855	0.6451	0.6800	-	-
**2019**					
year	0.7840	0.7067	0.6533	-	-
heating season	0.7882	0.7555	0.6573	-	-
non-heating season	0.7799	0.6624	0.497	-	-
**2020**					
year	0.8104	0.6900	0.6833	-	-
heating season	0.8327	0.7225	0.6938	-	-
non-heating season	0.7761	0.6523	0.6705	-	-
**2021**					
year	0.8063	0.8804	0.7178	0.7825	0.8066
heating season	0.8559	0.9877	0.7617	0.8577	0.8229
non-heating season	0.7634	0. 7557	0.6755	0.7158	0.7916

**Table 2 ijerph-20-00138-t002:** Spearman rank correlation coefficients between PM10 and selected meteorological parameters in 2018–2021.

	T_a_	u_a_	P_a_
PM10	−0.386 *	−0.614 *	0.164 *

T_a_—air temperature, u_a_—wind speed, P_a_—air pressure, * significance *p* < 0.001.

**Table 3 ijerph-20-00138-t003:** Comparisons of concentrations of PM and PM-bound B[a]P at various locations.

Location	Sampling Period	Sampling Point		Fraction PM	PM, µg·m^−3^	B[a]P, ng·m^−3^	Reference
Islamabad, Pakistan	2017	clean urban		PM2.5	51.59	1.55	[83]
				PM10	102.79	2.3	
Lahore, Pakistan		traffic		PM10	188.7	81.4	[84]
Amritsar, India	XI 2013–I 2014	urban		PM10	196–462	1.6	[85]
Raipur, India	XI 2013–I 2014	urban		PM10	468 (291–783)	43–76	[86]
Kolkata, India	X 2015–V 2016	urban		PM10	nd	5.3	[87]
Darjeeling, India						2.5	
Jamshedpur, India	XII 2016–V 2017	semi-urban	winter	PM2.5	nd	1.24	[88]
			summer			2.45	
		urban	winter			4.74	
			summer			3.42	
		rural	winter			1.95	
			summer			4.39	
Jorhat, India	II–III 2018	urban		PM2.5	126.8	-	[89]
				PM10	260.6	-	
	X–XI 2018			PM2.5	159.0	4.77	
				PM10	274.2	5.48	
Tangshan, China	2014	urban		PM2.5	23–367	3.64 * 61.6 **	[90]
Mlada Boleslav, Czech Republic	15–28 II 2013	residential district		PM1	26	0.78	[24]
Zagrzeb, Croatia	2014	urban	winter	PM1	nd	2.23	[91]
			spring			0.21	
			summer			0.03	
			autumn			0.59	
			winter	PM2.5		2.39	
			spring			0.27	
			summer			0.03	
			autumn			0.73	
			winter	PM10		7.66	
			spring			0.75	
			summer			0.07	
			autumn			1.35	
Sarajevo, Bosnia and Herzegovina	XII 2017–II 2018	urban background		PM10	50	7.28	[11]
Zagreb, Croatia	XII 2017–II 2018	residential			40	3.24	
Athenus, Greek	XII 2016–I 2018	urban background	winter	PM2.5	nd	0.81	[92]
			spring			0.02	
			summer			0.04	
			autumn			0.10	
Silesia, Poland	VIII 2009–XII 2010	regional background	non-heating	PM2.5	19.83	2.43	[23]
			heating		31.12	7.02	
		urban background	non-heating		23.80	2.89	
			heating		75.39	25.40	
		traffic	non-heating		27.97	5.65	
			heating		50.56	14.05	
Wadowice, Poland	III 2017	urban background		PM10	34.4 (10.8–116.7)	11.1	[93]
	VIII 2017					1.4	
Wadowice, Poland	II–X 2017	urban background	non-heating	PM10	27.1	1.1	[94]
			heating		43.3	4.98	
Silesia, Poland	2018	urban background		PM10	43.2	7.1	This study
	2019				35.1	6.3	
	2020				30.5	6.2	
	2021				32.3	6.3	

* non-heating period; ** heating period; nd—no data.

**Table 4 ijerph-20-00138-t004:** Spearman’s rank correlation coefficients between PM10 and B[a]P in 2018–2021.

	R Spearman	*t* (N−2)	*p*
Częstochowa	0.721	14.855	0.000
Katowice	0.723	15.042	0.000
Rybnik	0.843	22.531	0.000
Żywiec	0.873	25.545	0.000

R Spearman—correlation coefficient, *p*—significance level, *t*—significance test result for correlation coefficient.

**Table 5 ijerph-20-00138-t005:** Descriptive data on lifetime cancer risk (LCR) from the inhalation exposure to PM10-bound B[a]P in ambient air according to IRIS [57].

Air Monitoring Station	Annual	Heating Season	Non-Heating Season
Bielsko-Biała	2.36 × 10^−6^	4.05 × 10^−6^	0.32 × 10^−6^
Częstochowa	1.66 × 10^−6^	2.79 × 10^−6^	0.33 × 10^−6^
Godów	5.19 × 10^−6^	8.66 × 10^−6^	0.82 × 10^−6^
Katowice	2.38 × 10^−6^	4.15 × 10^−6^	0.42 × 10^−6^
Rybnik	6.89 × 10^−6^	11.95 × 10^−6^	1.08 × 10^−6^
Żywiec	4.53 × 10^−6^	8.00 × 10^−6^	0.62 × 10^−6^

**Table 6 ijerph-20-00138-t006:** Lifetime cancer risk and number of additional lung cancer cases due to exposure to long-term average (2018–2021) concentrations of B[a]P (ng·m^−3^) in the Silesian Voivodeship according to IRIS, EPA and WHO.

	Period	IRIS [57]	EPA [60]	WHO [61]
LCR	annual	3.88 × 10^−6^	7.12 × 10^−6^	56.33 × 10^−6^
	heating season	6.72 × 10^−6^	12.32 × 10^−6^	97.44 × 10^−6^
	non-heating season	0.60 × 10^−6^	1.10 × 10^−6^	8.70 × 10^−6^
Number of additional lung cancer cases	annual	17	31	248
	heating season	30	54	429
	non-heating season	3	5	38

## Data Availability

All data used in the paper are publicly available. We may supply the data that we gathered from public sources upon reasonable request to the corresponding author.

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
