# Peer review of "Spatial and Temporal Volatility of PM2.5, PM10 and PM10-Bound B[a]P Concentrations and Assessment of the Exposure of the Population of Silesia in 2018–2021"

_ijerph, 2022, doi:10.3390/ijerph20010138_

Round 1
Reviewer 1 Report
General comments:
In this study, a study on Spatial and temporal volatility of PM2.5, PM10, and PM10-bound B[a]P concentrations and assessment of the exposure is done, which has some research value. However, some major issues should be addressed before it can be considered for potential publication in the esteemed Journal.
Specific comment:
1. It is suggested that the results and discussion section of the paper be adjusted. The analysis of the relationship between PM2.5, PM10 and meteorological conditions is a very important part of the paper, so it is suggested that it be excluded as a separate subsection for analysis.
2. PM2.5 and PM10 are not only influenced by meteorological factors such as temperature, wind speed and air pressure, but also solar radiation, precipitation and humidity play an important role in particulate pollution. Why did the authors choose only temperature, wind speed and air pressure as the meteorological factors? Is it because of the data acquisition problem? If it is not a problem of data acquisition, then it is necessary to add the analysis of solar radiation, precipitation, and humidity on particulate matter.
3. There are more textual and formatting problems in the text, which are suggested to be seriously revised and improved, including but not limited to the following: for example, line P117 has obvious formatting errors; [57] in Table 6.
4. It is suggested to improve and standardize the figures in the paper, including but not limited to the following: (1) Figure 1, what the three types of areas in the middle represent respectively, need to be described in the figure name, so that others can easily understand the information through the figure without having to look up the description in the text. (2) Figure 2, it is suggested to improve it properly, it can be in the form of main map plus subsidiary map, in addition, add some physical geographic elements in the map, such as DEM, etc.
Author Response
Response to the comments of Reviewer 1
General comments:
In this study, a study on Spatial and temporal volatility of PM2.5, PM10, and PM10-bound B[a]P concentrations and assessment of the exposure is done, which has some research value. However, some major issues should be addressed before it can be considered for potential publication in the esteemed Journal.
Specific comment:
- It is suggested that the results and discussion section of the paper be adjusted. The analysis of the relationship between PM2.5, PM10 and meteorological conditions is a very important part of the paper, so it is suggested that it be excluded as a separate subsection for analysis.
As suggested by the reviewer, we have added an additional subchapter: 3.2 Relationship between some meteorological parameters and particulate matter concentrations.
- PM2.5 and PM10 are not only influenced by meteorological factors such as temperature, wind speed and air pressure, but also solar radiation, precipitation and humidity play an important role in particulate pollution. Why did the authors choose only temperature, wind speed and air pressure as the meteorological factors? Is it because of the data acquisition problem? If it is not a problem of data acquisition, then it is necessary to add the analysis of solar radiation, precipitation, and humidity on particulate matter.
We are aware that PM10 concentrations are affected by many meteorological factors that significantly affect the diffusion, deposition or dilution of PM10. However, due to data availability problems, we were unable to analyze all of them. Only temperature, wind speed and direction, and air pressure were accessible at the Rybnik monitoring station.
- There are more textual and formatting problems in the text, which are suggested to be seriously revised and improved, including but not limited to the following: for example, line P117 has obvious formatting errors; [57] in Table 6.
We had a problem with the text formatting indeed. All mistakes spotted in the manuscript have been seriously revised and improved.
- It is suggested to improve and standardize the figures in the paper, including but not limited to the following: (1) Figure 1, what the three types of areas in the middle represent respectively, need to be described in the figure name, so that others can easily understand the information through the figure without having to look up the description in the text. (2) Figure 2, it is suggested to improve it properly, it can be in the form of main map plus subsidiary map, in addition, add some physical geographic elements in the map, such as DEM, etc.
Figures 1 and 2 have been corrected in accordance with your recommendations

Reviewer 2 Report
The manuscript contributes with PM and BaP data. The discussion is well supported. A few suggestions can be found in the text. I would recommend improving figures I and 2 (the maps are quite old). I strongly suggest including the methodology of collection and information on the measurements BaP concentration and the standard(s) used. I think that once revised the manuscript with the observations made it, the manuscript will have the quality to be published at the IJERPH.

Author Response
Response to the comments of Reviewer 2
The manuscript contributes with PM and BaP data. The discussion is well supported. A few suggestions can be found in the text. I would recommend improving figures I and 2 (the maps are quite old). I strongly suggest including the methodology of collection and information on the measurements BaP concentration and the standard(s) used. I think that once revised the manuscript with the observations made it, the manuscript will have the quality to be published at the IJERPH.
In accordance with your recommendations:
- figures 1 and 2 have been corrected
- the information on PM and B[a]P measurements methodology was added. Chapter 2 has been also slightly reorganized.

Round 2
Reviewer 1 Report
The article has been improved to some extent and is recommended to be accepted after further improvement of the cartography, text, reference format and grammar.